# Occupational Branch and Labor Market Marginalization among Young Employees with Adult Onset of Attention Deficit Hyperactivity Disorder—A Population-Based Matched Cohort Study

**DOI:** 10.3390/ijerph19127254

**Published:** 2022-06-14

**Authors:** Katalin Gémes, Emma Björkenstam, Syed Rahman, Klas Gustafsson, Heidi Taipale, Antti Tanskanen, Lisa Ekselius, Ellenor Mittendorfer-Rutz, Magnus Helgesson

**Affiliations:** 1Department of Clinical Neuroscience, Division of Insurance Medicine, Karolinska Institutet, 17177 Stockholm, Sweden; emma.bjorkenstam@ki.se (E.B.); syed.rahman@ki.se (S.R.); klas.gustafsson@ki.se (K.G.); heidi.taipale@ki.se (H.T.); ellenor.mittendorfer-rutz@ki.se (E.M.-R.); magnus.helgesson@ki.se (M.H.); 2Department of Medical Sciences, Uppsala University, 75185 Uppsala, Sweden; 3Niuvanniemi Hospital, FI-70240 Kuopio, Finland; antti.tanskanen@ki.se; 4School of Pharmacy, University of Eastern Finland, FI-70211 Kuopio, Finland; 5Department Women’s and Children’s Health, Uppsala University, 75237 Uppsala, Sweden; lisa.ekselius@neuro.uu.se

**Keywords:** attention deficit hyperactivity disorder (ADHD), labor market marginalization, sickness absence, unemployment, occupational branches, young adults

## Abstract

We compared labor market marginalization (LMM), conceptualized as days of unemployment, sickness absence and disability pension, across occupational branches (manufacturing, construction, trade, finance, health and social care, and education), among young employees with or without attention deficit hyperactivity disorder (ADHD) and examined whether sociodemographic and health-related factors explain these associations. All Swedish residents aged 19–29 years and employed between 1 January 2005 and 31 December 2011 were eligible. Individuals with a first ADHD diagnosis (*n* = 6030) were matched with ten controls and followed for five years. Zero-inflated negative binomial regression was used to model days of LMM with adjustments for sociodemographic and health-related factors. In total, 20% of those with ADHD and 59% of those without had no days of LMM during the follow-up. The median of those with LMM days with and without ADHD was 312 and 98 days. Having an ADHD diagnosis was associated with a higher incidence of LMM days (incident rate ratios (IRRs) 2.7–3.1) with no differences across occupational branches. Adjustments for sociodemographic and health-related factors explained most of the differences (IRRs: 1.4–1.7). In conclusion, young, employed adults with ADHD had a higher incidence of LMM days than those without, but there were no substantial differences between branches, even after adjusting for sociodemographic and health-related factors.

## 1. Introduction

Attention deficit hyperactivity disorder (ADHD) is a neurodevelopmental disorder usually diagnosed during childhood, but from the 2000s it has more widely been recognized among adults, the working-age population [1,2,3]. The prevalence of ADHD among adults in high-income countries is estimated to be around 3.6% [4], and the number of diagnosed cases among young adults has been increasing [5,6,7]. Symptoms often include attention deficiencies, problems with controlling activity levels and impulsiveness, which might influence social integration, educational attainment and work ability negatively [8,9,10,11,12,13].

There have been relatively few studies so far that have investigated whether being diagnosed with ADHD in young adulthood influences a sustainable working life. Previous studies on young adults with mental disorders reported that many have problems of entering and being able to stay in the labor market [9,10,13]. Young adults with ADHD have more sickness absence (SA) days if working and a higher risk of being on disability benefit or unemployed compared with young adults without ADHD [11,13,14,15,16,17]. Lower socioeconomic status such as a low level of education, living in more rural settings [14,17] and having comorbid mental disorders [14,15,16] predicts a higher risk of work disability among individuals with ADHD diagnosed in adulthood. Being a woman and previous long-term SA is also identified as predictors for long-term work disability among young individuals with ADHD [14]. However, it is not known whether there is any difference among occupational branches, such as manufacturing, construction, trade and communication, financial and business services, education and research, and health and social work, in relation to adverse labor market outcomes for young employees who have entered the labor market and who have been diagnosed with ADHD during young adulthood. Occupational branches might differ to a large extent regarding work-related factors, such as the number and intensity of contacts with other people or psychosocial strain, which might lead to a challenging environment for employees with ADHD and increase the risk of labor market marginalization (LMM). According to the “core-peripheral” theoretical framework of labor market integration, there is a continuum from attachment to marginalization [18,19]. Our previous research show that a large proportion of young adults with an ADHD diagnosis are already on the periphery of the labor market and were marginalized early [12]. This study aims to understand to what extent labor market-related factors, such as the occupational branch, might play a role in LMM among young individuals with ADHD who have managed to establish labor market attachment. The availability of register data on the total population provides an opportunity to examine this question across the whole population of Sweden.

Therefore, by using nationwide register information, we aimed to investigate the association between ADHD diagnosis in adulthood and LMM, conceptualized from a social insurance perspective as unemployment, SA and disability pension (DP) [20] through the occupational branches. Furthermore, we also investigated whether sociodemographic and health-related factors can explain the associations between having ADHD and LMM for these individuals within the different occupational branches.

## 2. Materials and Methods

### 2.1. Study Population

We conducted a prospective register-based matched cohort study. We followed the STROBE guideline/checklist for reporting the study [21] (Appendix A). All individuals between the ages of 19 and 29 years old, resident in Sweden between January 2006 and December 2011, and employed (including self-employed) during the year before the study entry were eligible for our study. We identified individuals with a first diagnosis of ADHD, defined by the code F90, according to the International Classification of Diseases, 10th Revision during this period (ICD-10) [22], and who had a health care record with this diagnosis in specialized health care. In Sweden, ADHD that has severe enough symptoms to affect life quality is diagnosed exclusively in specialized healthcare centers using standardized interview [23]. To increase statistical efficacy, for each of the ADHD cases, we matched ten individuals without a history of ADHD, on age, sex, occupational branch and year of cohort entry (i.e., the year of the diagnosis). Individuals were followed for five years after the study entry. From the 66,330 participants, 151 emigrated (defined as disappearing from the population register during the follow-up, but not registered on the Cause of Death Register) and 1345 died (identified from the Cause of Death register) during the follow-up (the percentage (2%) was the same among individuals with and without a diagnosis of ADHD).

### 2.2. Data Source

All information used in this study was obtained as microdata from the following national registers and linked at an individual level through the unique personal identification number [24]:

The Longitudinal Integration Database for Health Insurance and Labor Market Studies (LISA), held by Statistics Sweden, contains information on sociodemographics, such as sex, year of birth, year of emigration, type of living area, country of birth, family composition, educational level, labor market participation and other work-related information such as occupational branch, income and unemployment [25]. The National Patient Register, held by the National Board of Health and Welfare was used to identify diagnosis of ADHD and comorbid disorders based on ICD-10 codes. The Cause of Death register was used to identify date of death during the study period [26]. Information on SA >14 days and DP including date, duration grade and main diagnosis were obtained from the Microdata for Analyses of Social Insurance (MiDAS), from the Swedish Social Insurance Agency [27].

### 2.3. Study Variables

*Outcomes:* labor market marginalization (LMM) days were measured as work disability, i.e., the sum of net days with SA, DP, or days of full-term unemployment during the follow-up period, in order to capture both medically certified and non-medically certified absence from work. Employees in Sweden are eligible for paying SA for the first 14 days of the sickness spell; a medical certificate is required from day 8. Time-restricted DP can be granted to any individuals between the ages of 19 and 29 whose work capacity/ability has been reduced by at least 25%. Temporary DP can become permanent after the age of 30 if the same conditions persist. Both SA and DP can be granted in part-time and full-time [27] amounts. In the main analysis we accounted for the annual net days of SA, DP and unemployment, and the sum of them was calculated as LMM days.

In secondary analysis, we also investigated the sum of SA/DP days and unemployment days separately. Unemployment days were calculated on an annual basis. SA/DP days were available on a daily basis; therefore, we could calculate this outcome from the exact cohort entry date (i.e., 13 March 2005–12 March 2010).

*Predictors:* the detailed categorization of the predictors is presented in Table 1 by ADHD diagnosis. Occupational sector codes were defined by Statistic Sweden based on the companies’ institutional sector code (Swedish Standard Classification of Occupations—www.scb.se, accessed on 5 June 2022) and were further categorized to the following wider occupational branches: (1) manufacturing, (2) construction, (3) trade and communication, (4) financial and business services, (5) education and research, (6) health and social care, (7) other (including agriculture, forestry, fishing, energy production, water supply and waste management, personal and cultural services, public administration and non-specified services). Sex (male; female), age, educational level (low (<10); medium (10–12); high (>12 years)), country of origin (Sweden; other Nordic countries; other European Union 27 countries; rest of the world), family composition (married/cohabiting or single with children or without children), type of living area (large cities; medium size cities; small towns/villages), blue- or white-collar job and income in quartiles were considered as sociodemographic variables. The following health-related variables were considered: History of SA was defined as any or no. The following diagnoses (based on ICD-10 codes) were considered as comorbidities: depression and bipolar disorders (ICD-10: F30–F34), anxiety- and stress-related disorders (ICD-10: F40–F48), autism spectrum disorders (ICD-10: F84), substance use (ICD-10: F10–F19), behavioral and emotional disorders (ICD-10: F91–F98), schizophrenia/non-affective psychoses (ICD-10: F20–F29), other mental disorders (ICD-10 other F-codes), musculoskeletal disorders (ICD-10: M01–M99) and other somatic disorders (all other ICD-10 except O.80 and Z00-99).

### 2.4. Statistical Analysis

Differences in sociodemographic and health-related factors between individuals with ADHD and without ADHD were summarized and compared by calculating proportions and chi^2^ statistics. As LMM days is a non-negative integer value with zero-inflated distribution, therefore, in descriptive statistics we presented the percentage of individuals with no LMM days during the follow-up, and the median and interquartile range (IQR) among those who had at least one LMM day during the follow-up. As conditional variance of the outcome exceeded its means, resulting in overdispersion and as there were excessive zero counts as most of the individuals had no LMM days, we used zero-inflated negative binomial regressions (zinb) to model the association between ADHD diagnosis and LMM [28]. Zero-inflated binomial regression allows for excess zeros to be modelled independently, using logistic regression and estimates of odds ratios (ORs) of not being at risk of LMM (having zero LMM days). For those who had LMM days, a negative binomial regression was used to estimate the incidence rate ratios (IRRs) of LMM days during the follow-up. The counts of LMM days were modelled in relation (offset) to the total number of days the individuals were followed, from cohort entry until death/emigration or until the end of the 5-year follow-up period [28] and IRRs and 95% confidence intervals (CIs) were calculated to compare the IRRs of LMM days among those with and without ADHD. The reciprocals of ORs of zero LMM days and IRRs of LMM days were presented for the respective models along with 95% CIs.

Four models, with adjustments for different sets of covariates, were run. Model 1 was only adjusted for age, model 2 was further adjusted for sociodemographic factors, model 3 was additionally adjusted for previous SA and somatic comorbidities and model 4 further for mental disorders. In a secondary analysis we ran the same models separately for the sum of SA/DP days and unemployment days. We also ran model 4 for the whole study sample twice, once without and once with adjustment for occupational branches as a categorical covariate, to examine how much differences between occupational branches explained the association between ADHD diagnosis and LMM. Finally, we stratified the analyses by individuals with a diagnosis of ADHD with and without any mental comorbidity while adjusting to occupational branches and other covariates.

Data management was performed with SAS Base (version: 9.4, SAS Institute AB, Cary, NC, USA) and statistical analysis with R (version 4.0.5, R Foundation for Statistical Computing, Vienna, Austria) using the “zeroinfl” function from the “pscl” package. The Regional Ethical Review Board of Stockholm, Sweden, approved our study.

## 3. Results

### 3.1. Descriptive Statistics

Table 1 presents the distribution of the study variables, in total and by ADHD diagnosis. The median age for people diagnosed with ADHD in the cohort was 24 years. Those with ADHD were more likely to have low educational level, be born in Sweden, have lower income, work in blue-collar jobs and were more often diagnosed with mental disorders compared with individuals without ADHD. Overall, 19% of those with ADHD and 59% of those without had no LMM days during the follow-up. Among those who had LMM days, the median of LMM days was 312 (IQR: 128, 738) and 98 (IQR: 42, 219) days among those without. Individuals with ADHD had a substantially higher prevalence of at least one LMM day and higher median of LMM days across all branches, with the lowest prevalence of zero LMM days observed among individuals both with and without ADHD in the health and social care branch: 15% and 49%, respectively (Appendix A).

**Table 1 ijerph-19-07254-t001:** Distribution of study variables by attention deficit hyperactivity disorder (ADHD) diagnosis among young employed individuals.

Study Variables	Total *n* = 66,330	With ADHD *n* = 6030	Without ADHD *n* = 60,300	*p*-Values for Chi² Test for Independence
**Age, mean (SD)**	24.3 (3.0)	24.2 (3.0)	24.2 (3.0)	
	*N* (%)	*n* (%)	*n* (%)	
** *Sex* **				
**Female**	30,107 (45.4)	2737 (45.4)	27,370 (45.4)	
**Male**	36,223 (54.6)	3293 (54.6)	32,930 (54.6)	
** *Educational level* **				<0.01
**Low (<10 years)**	7229 (10.9)	1783 (29.6)	5446 (9.0)	
**Medium (10–12 years)**	41,196 (62.1)	3547 (58.8)	37,649 (62.4)	
**High (>12 years)**	17,905 (27.0)	700 (11.6)	17,205 (28.5)	
** *Country of origin* **				<0.01
**Sweden**	58,857 (88.7)	5640 (93.5)	53,217 (88.3)	
**Nordic countries**	397 (0.6)	38 (0.6)	359 (0.6)	
**Other European country**	1237 (1.9)	54 (0.9)	1183 (2.0)	
**Other**	5839 (8.8)	298 (4.9)	5541 (9.2)	
** *Family composition* **				<0.01
**Married/cohabiting without children**	1970 (3.0)	98 (1.6)	1872 (3.1)	
**Married or cohabitant with children**	8657 (13.1)	706 (11.7)	7951 (13.2)	
**Single without children**	54,456 (82.1)	4908 (81.4)	49,548 (82.2)	
**Single with children**	1247 (1.9)	318 (5.3)	929 (1.5)	
** *Type of living area* **				<0.01
**Large city**	26,341 (39.7)	2714 (45.0)	23,627 (39.2)	
**Medium-sized city**	23,553 (35.5)	1802 (29.9)	21,751 (36.1)	
**Small town/village**	16,436 (24.8)	1514 (25.1)	14,922 (24.7)	
** *Income quantile* **				<0.01
**First**	16,601 (25.0)	2076 (34.4)	14,525 (24.1)	
**Second**	16,573 (25.0)	1896 (31.4)	14,677 (24.3)	
**Third**	16,589 (25.0)	1309 (21.7)	15,280 (25.3)	
**Fourth**	16,567 (25.0)	749 (12.4)	15,818 (26.2)	
** *Blue- or white-collar worker* **				<0.01
**Blue-collar**	43,886 (76.5)	4364 (87.1)	39,522 (75.4)	
**White-collar**	13,507 (23.5)	646 (12.9)	12,861 (24.6)	
** *Type of work* **				0.69
**Employed**	64,312 (97.0)	5841 (96.9)	58,471 (97.0)	
**Self-employed**	2018 (3.0)	189 (3.1)	1829 (3.0)	
** *Sickness absence one year prior to baseline* **				<0.01
**Yes**	5813 (8.8)	1969 (32.7)	3844 (6.4)	
**No**	60,517 (91.2)	4061 (67.3)	56,456 (93.6)	
** *Mental disorders* **				
**Depression/bipolar disorders**	2837 (4.3)	2056 (34.1)	781 (1.3)	<0.01
**Anxiety/stress-related disorders**	3330 (5.0)	2267 (37.6)	1063 (1.8)	<0.01
**Autism spectrum disorders**	436 (0.7)	415 (6.9)	21 (0.0)	<0.01
**Substance abuse**	1650 (2.5)	1176 (19.5)	474 (0.8)	<0.01
**Behavioural/emotional disorders**	253 (0.4)	241 (4.0)	12 (0.0)	<0.01
**Schizophrenia/psychoses**	160 (0.2)	102 (1.7)	58 (0.1)	<0.01
**Other mental disorders**	1439 (2.2)	1094 (18.1)	345 (0.6)	<0.01
** *Somatic disorders* **				
**Musculoskeletal disorders**	4258 (6.4)	681 (11.3)	3577 (5.9)	<0.01
**Other somatic disorders**	29,415 (44.6)	3940 (65.3)	15,475 (42.2)	<0.01
**LMM * days**				
**No LMM days during follow-up, *n* (%)**	35,818 (54%)	1206 (20%)	35,577 (59%)	
**Median (IQR) of LMM days among >0 days**	114 (48, 272)	312 (128, 738)	98 (42, 219)	

SD: standard deviation. LMM: labor market marginalization. IQR: interquartile range. * Measured as the summary of sickness absence, disability pension and unemployment net days during the study period.

### 3.2. Main Analysis

A diagnosis of ADHD was associated with a higher probability of LMM in all occupational branches (Table 2). The highest ORs were in model 1, observed among those who were employed within the “finance and business service” and “manufacturing” branches, and in the full adjusted model in the “trade and communication” and “finance and business service” occupational branches (Table 2). The ORs decreased slightly after adjustments were made for sociodemographic factors as well as for somatic disorders, and decreased further in all branches after additional adjustment for other mental disorders.

The IRRs of LMM days were almost three times higher among individuals with diagnosed ADHD compared with individuals without the ADHD diagnosis in all occupational branches. Adjusting for sociodemographic variables and somatic disorders explained part of the observed differences. After adjusting for mental disorders, the IRRs decreased to 1.3–1.6 throughout the branches (Table 2). The fully adjusted IRR was the lowest in the “education and research” branch, and the highest in the “trade and communication” and “financial and business services” branches: adjusted IRRs 1.3 (95% CI: 1.2, 1.6), 1.6 (95% CI: 1.4, 1.7) and 1.6 (95% CI: 1.4, 1.8), respectively.

### 3.3. Secondary Analysis

The ORs of experiencing SA/DP when compared with those with and without ADHD diagnosis were highest in the “financial and business service” branch and lowest in the “health and social care” branch in the crude models (Table 3). Adjusting for sociodemographic and health factors did not explain the differences between individuals with and without ADHD but decreased them substantially. The estimates were similar across branches. The IRR estimates followed similar patterns. We observed the highest IRR in the “financial and business service” branch between individuals with and without ADHD (adjusted IRR: 2.1, 95% CI: 1.8, 2.5) and the lowest in the “education and research” branch (adjusted IRRs: 1.4, 95% CI: 1.1, 1.7). Regarding unemployment days, the crude ORs and IRRs showed similar patterns as the other outcomes (Table 4). Both the ORs and the IRR estimates decreased substantially when the analysis was adjusted for all the covariates.

The estimates for LMM comparing those with and without ADHD diagnosis did not change when occupational branches were included into the fully adjusted model as a covariate (Appendix A). The results comparing individuals with ADHD diagnosis with and without any mental comorbidities (*n* = 4045 and *n* = 1985, respectively) are presented in Appendix A. The probability of LMM was higher among those who had both ADHD and a mental comorbidity (OR 4.5, 95% CI: 4.1, 5.0) than those who were diagnosed with ADHD without other mental comorbidities (OR 2.3, 95% CI: 2.0, 2.5) and compared to the reference group free from ADHD. However, the associations between the different occupational branches were similar in the two subgroups of ADHD (with and without mental comorbidities) (Appendix A).

## 4. Discussion

### 4.1. Summary of Findings

Young employed adults diagnosed with ADHD had a higher risk of LMM, both as a higher number of unemployment days and SA/DP, compared with their peers without ADHD in all occupational branches. These differences were to a large extent explained by sociodemographic and health-related factors. There were no substantial differences between the occupational branches concerning the risk of LMM. However, it has to be mentioned that in health and social care, the absolute difference of LMM days, especially SA/DP days, was large between individuals with and without ADHD.

The similar relative differences in the risk of LMM between individuals with and without ADHD through occupational branches were in contradiction with our hypothesis that in occupational branches including occupations with high numbers of contacts with people the risk of LMM among individuals with ADHD is higher compared with other branches. This lack of relative difference might be due to the fact that individuals with ADHD to some extent adjust their occupational choices according to their ADHD symptoms and self-select occupations and occupational branches that fit better for their workability [29]. Observing the unemployment and SA/DP outcomes separately showed a similar pattern: no substantial differences could be observed between the occupational branches. However, it is worth mentioning that there were substantial differences in the absolute number of LMM days between the branches, with the highest number of LMM days in the health and social care branches among young individuals both with and without ADHD. Work disability, especially due to mental disorders, is a well-described problem in these branches, among others, due to work characteristics such as high job strain, effort–reward imbalances, and low job security [30,31,32,33,34,35,36]. This suggests that the work environment that is specific for occupational branches affects individuals with and without ADHD to a similar extent.

Several studies have reported that individuals with ADHD diagnosed in childhood and young adulthood have difficulties completing their education [37], as well as entering and staying in the labor market [12,14,17,37,38,39]. In this study, individuals were already employed when diagnosed with ADHD [14], which indicates that they might have been in better health, and had less debilitating ADHD symptoms or better responsiveness to treatment compared with ADHD patients on permanent DP [12]. However, even in this highly selected group, individuals with ADHD still had a higher risk of LMM compared with those without. These differences were explained to a large degree by differences in sociodemographic and health-related factors, especially by other mental comorbidities in all occupational branches. A large heterogeneity of individuals diagnosed by ADHD, with different degrees of comorbidities and work-related factors, has been described previously [17,39,40]. In general ADHD patients with a lower socioeconomic status, such as lower level of education, lower income, living in rural areas and with common mental disorders, had a higher risk of DP and unemployment compared with ADHD patients with higher socioeconomic status and without mental comorbidities [14,17,39]. Comorbid mental disorders might contribute to worse LMM by impairing adherence to treatment [40,41], but they can also directly affect workability [42,43,44]. Our results suggest that preventing or treating comorbid mental disorders among persons with ADHD might have an important implication in decreasing consequent LMM regardless of occupational branches [40,45]. It has to be mentioned that intervention in a supportive and modified work environment, increased employers’ knowledge and awareness, as well as integrated comprehensive career-focused approaches in treatment are ways to improve the work ability of individuals with ADHD [11,29,43], but we found no indication of major differences regarding LMM among occupational branches. However, as the absolute difference in LMM days, especially SA, are considerable in the “health and social care” occupational branch, these branch interventions should focus on both targeting the general work environment as well as individuals with special needs.

### 4.2. Strengths and Limitations

One of the main strengths of our study was that we used high quality, longitudinal register data of the whole population of Sweden, including a wide-range of sociodemographic and health-related covariates [25,46]. The study population covered all young individuals who were employed during the study period in Sweden, and the loss of individuals during the follow-up was minimal. We used the sum of SA/DP and unemployment net days to conceptualize LMM and avoid underestimation, which can be a problem if using only one of the measures individually [47]. Furthermore, using a broad definition of LMM including both work disability and unemployment increases comparability with other countries with different welfare systems [20]. In addition, work disability and unemployment were analysed separately to ensure that there were no significant subgroup effects and that the results were in line with the main analysis.

One of the limitations of our study is that ADHD and comorbid disorders were captured by visits at specialized outpatient and inpatient health care facilities; therefore, conditions that were only treated in primary care might have been missed. However, in Sweden, ADHD that has severe enough symptoms to influence education, working life, social behaviour or other aspects of life is exclusively diagnosed and mainly treated in specialized healthcare centers, even if the first contact due to the symptoms may have been with primary care [23]. Another limitation is that we only had information on SA spells longer than 14 days as the first two weeks are covered by the employers and not registered by the Swedish Social Insurance Agency. This might lead to some slight underestimation of the absolute difference in LMM days. If individuals with ADHD are more likely to have had short SA < 15 days compared with individuals without ADHD, this could also have led to a slight underestimation of the relative differences, but it is unlikely that it would affect our conclusions. Finally, while information on employment was referred to the year before cohort entry to ensure that all study participants were employed before their ADHD diagnosis, we cannot rule out that the symptoms of ADHD had already influenced the participants’ working life before the diagnosis.

As our study only focused on outcomes among individuals who already had some attachment to the labor market, the results cannot be applied to all individuals diagnosed with ADHD. Often adults with high IQ and probably a supportive environment might function well in elementary or secondary education, and ADHD will first become a barrier when they reach higher level education or meet professional demands and are not able to further compensate for their symptoms [29]. Individuals with more serious symptoms, with lower cognitive abilities or less supportive environment might not be able to enter the labor market [12,14].

## 5. Conclusions

There were no substantial differences between occupational branches concerning LMM between young employed adults with and without ADHD. Differences in sociodemographic and health-related factors, especially comorbid mental disorders, accounted for a large part of the differences in LMM days similarly across all occupational branches. In some occupational branches the absolute difference in LMM days between individuals with and without ADHD was substantial. Focusing on the treatment of comorbid mental disorders, education and adjustments in the work environment might target interventions to prevent LMM in individuals with ADHD. 

## Figures and Tables

**Table 2 ijerph-19-07254-t002:** Labor market marginalization (LMM) by occupational branches among young employed individuals with and without attention deficit hyperactivity disorder (ADHD).

	Manufacturing	Construction	Trade and Communication	Financial and Business Service	Education and Research	Health and Social Care	Other
Odds ratios and 95% confidence intervals of being at risk of LMM * days during the study period (reference category: individuals without ADHD)							
Model 1	6.7 (5.3, 8.3)	6.3 (5.0, 7.7)	5.9 (5.0, 6.7)	6.7 (5.6, 8.3)	5.6 (4.2, 7.1)	5.3 (4.6, 6.3)	5.6 (4.8, 6.7)
Model 2	5.0 (4.0, 6.3)	5.0 (4.0, 6.3)	5.0 (4.4, 5.6)	5.0 (4.0, 6.3)	4.8 (3.5, 6.3)	4.4 (3.7, 5.0)	4.8 (4.2, 5.6)
Model 3	3.5 (2.8, 4.4)	3.5 (2.8, 4.4)	3.5 (2.9, 4.0)	3.3 (2.6, 4.2)	3.5 (2.5, 4.6)	2.9 (2.5, 3.6)	3.5 (2.9, 4.0)
Model 4	2.4 (1.8, 3.1)	2.5 (1.9, 3.2)	2.6 (2.2, 3.1)	2.6 (2.0, 3.5)	2.3 (1.6, 3.3)	2.0 (1.7, 2.4)	2.2 (1.8, 2.6)
**Incidence rate ratios and 95% confidence intervals of LMM days (reference category: individuals without ADHD)**							
Model 1	2.9 (2.6, 3.2)	2.7 (2.5, 3.0)	2.9 (2.7, 3.1)	2.7 (2.4, 3.0)	2.7 (2.4, 3.1)	3.1 (2.8, 3.3)	2.7 (2.5, 2.9)
Model 2	2.5 (2.3, 2.7)	2.5 (2.2, 2.7)	2.7 (2.5, 2.8)	2.5 (2.3, 2.8)	2.3 (2.0, 2.6)	2.7 (2.5, 2.9)	2.4 (2.3, 2.6)
Model 3	2.1 (2.0, 2.4)	2.1 (1.9, 2.3)	2.2 (2.1, 2.4)	2.2 (2.0, 2.4)	1.8 (1.6, 2.1)	2.2 (2.1, 2.4)	2.0 (1.9, 2.2)
Model 4	1.5 (1.3, 1.6)	1.5 (1.4, 1.7)	1.6 (1.4, 1.7)	1.6 (1.4, 1.8)	1.3 (1.1, 1.5)	1.5 (1.4, 1.6)	1.4 (1.3, 1.5)

* LMM was defined as the sum of annual net unemployment, sickness absence and/or disability pension days. Model 1 was adjusted for age. Model 2 was further adjusted for years of education, country of origin, family status, living region, income, type of employment, blue/white-collar job. Model 3 was further adjusted for somatic disorders and having sickness absence prior to the study entry. Model 4 was further adjusted for other mental disorders. Zero-inflated binomial regression analysis was used to estimate the odds of not being at risk of the outcome and the count of the outcomes during the follow-up. The reciprocal values of the odds ratios and 95% confidence interval are presented in the table.

**Table 3 ijerph-19-07254-t003:** Sickness absence and disability pension (SA/DP) days by occupational branches among young employed individuals with and without attention deficit hyperactivity disorder (ADHD).

	Manufacturing	Construction	Trade and Communication	Financial and Business Service	Education and Research	Health and Social Care	Other
Odds ratios and 95% confidence intervals of being at risk of SA/DP during the study period (reference category: individuals without ADHD)							
Model 1	6.3 (5.3, 7.1)	5.0 (4.2, 5.9)	5.0 (4.4, 5.6)	6.7 (5.6, 7.7)	4.7 (3.6, 5.9)	4.6 (4.0, 5.3)	4.8 (4.2, 5.6)
Model 2	5.3 (4.6, 6.3)	4.4 (3.6, 5.0)	4.6 (4.2, 5.3)	5.6 (4.6, 6.7)	4.0 (3.1, 5.3)	4.2 (3.6, 4.7)	4.6 (3.9, 5.0)
Model 3	3.6 (2.9, 4.4)	2.9 (2.4, 3.5)	3.2 (2.8, 3.6)	3.6 (2.9, 4.4)	2.9 (2.2, 3.9)	2.9 (2.5, 3.3)	3.0 (2.6, 3.5)
Model 4	2.0 (1.6, 2.5)	1.8 (1.5, 2.3)	2.1 (1.8, 2.4)	2.1 (1.7, 2.8)	1.9 (1.3, 2.6)	1.8 (1.5, 2.2)	1.9 (1.6, 2.3)
**Incidence rate ratios and 95% confidence intervals (reference category: individuals without ADHD)**							
Model 1	4.3 (3.9, 4.9)	3.6 (3.2, 4.1)	3.8 (3.5, 4.1)	4.2 (3.6, 4.8)	3.0 (2.5, 3.6)	3.7 (3.4, 4.0)	3.6 (3.2, 3.9)
Model 2	3.3 (2.9, 3.8)	3.3 (2.9, 3.7)	3.3 (3.0, 3.6)	3.8 (3.3, 4.3)	2.6 (2.1, 3.0)	3.3 (3.0, 3.5)	3.0 (2.7, 3.3)
Model 3	2.8 (2.4, 3.2)	2.8 (2.5, 3.2)	2.7 (2.5, 3.0)	3.1 (2.7, 3.6)	2.0 (1.7, 2.4)	2.7 (2.5, 3.0)	2.4 (2.2, 2.7)
Model 4	1.7 (1.5, 2.0)	1.9 (1.6, 2.2)	1.8 (1.6, 2.0)	2.1 (1.8, 2.5)	1.4 (1.1, 1.7)	1.7 (1.5, 1.9)	1.5 (1.3, 1.7)

Model 1 was adjusted for age. Model 2 was further adjusted for years of education, country of origin, family status, living region, income, type of employment, blue/white-collar job. Model 3 was further adjusted for somatic disorders and having SA prior to the study entry. Model 4 was further adjusted for other mental disorders. Zero-inflated binomial regression analysis was used to estimate the odds of not being at risk of the outcome and the count of the outcomes during the follow-up. The reciprocal values of the odds ratios and 95% confidence interval are presented in the table.

**Table 4 ijerph-19-07254-t004:** Unemployment (UE) days by occupational branches among young employed individuals with and without attention deficit hyperactivity disorder (ADHD).

	Manufacturing	Construction	Trade and Communication	Financial and Business Service	Education and Research	Health and Social Care	Other
Odds ratios and 95% confidence intervals of being at risk of unemployment during the study period (reference category: individuals without ADHD)							
Model 1	3.7 (3.2, 4.4)	4.4 (3.7, 5.3)	4.0 (3.6, 4.6)	3.9 (3.3, 4.8)	3.7 (2.9, 4.8)	3.2 (2.9, 3.6)	3.7 (3.2, 4.2)
Model 2	2.9 (2.4, 3.5)	3.5 (2.9, 4.2)	3.3 (2.9, 3.7)	2.7 (2.2, 3.3)	3.0 (2.4, 4.0)	2.4 (2.1, 2.8)	2.9 (2.6, 3.5)
Model 3	2.6 (2.2, 3.1)	3.1 (2.6, 3.9)	2.9 (2.6, 3.3)	2.4 (1.9, 2.9)	2.8 (2.1, 3.6)	2.2 (1.9, 2.5)	2.6 (2.3, 3.0)
Model 4	2.0 (1.6, 2.5)	2.7 (2.2, 3.5)	2.6 (2.2, 3.0)	2.3 (1.8, 2.9)	2.5 (1.8, 3.3)	1.8 (1.5, 2.1)	2.1 (1.8, 2.4)
**Incidence rate ratios and 95% confidence intervals (reference category: individuals without ADHD)**							
Model 1	1.3 (1.1, 1.4)	1.2 (1.1, 1.4)	1.2 (1.1, 1.3)	1.1 (1.0, 1.3)	1.2 (1.0, 1.4)	1.0 (0.9, 1.1)	1.2 (1.1, 1.3)
Model 2	1.2 (1.1, 1.4)	1.1 (1.0, 1.3)	1.2 (1.1, 1.3)	1.1 (1.0, 1.2)	1.1 (1.0, 1.4)	1.0 (0.9, 1.1)	1.2 (1.1, 1.3)
Model 3	1.2 (1.1, 1.4)	1.1 (1.0, 1.3)	1.2 (1.1, 1.3)	1.1 (1.0, 1.3)	1.1 (0.9, 1.4)	1.0 (0.9, 1.1)	1.2 (1.1, 1.3)
Model 4	1.1 (1.0, 1.3)	1.1 (1.0, 1.3)	1.2 (1.1, 1.3)	1.1 (1.0, 1.3)	1.2 (0.9, 1.5)	1.1 (0.9, 1.2)	1.2 (1.0, 1.3)

Model 1 was adjusted for age. Model 2 was further adjusted for sex, years of education, country of origin, family status, living region, income, type of employment, blue/white-collar job. Model 3 was further adjusted for somatic disorders and having sickness absence prior to the study entry. Model 4 was further adjusted for diagnosis of other mental disorders. Zero-inflated binomial regression analysis was used to estimate the odds of not being at risk of the outcome and the count of the outcomes during the follow-up. The reciprocal values of the odds ratios and 95% confidence interval are presented in the table.

## Data Availability

The project utilized data from the REWHARD consortium, supported by the Swedish Research Council (VR grant number. 2017-00624). These data cannot be made publicly available due to privacy regulations. According to the General Data Protection Regulation, the Swedish law SFS 2018:218, the Swedish Data Protection Act, the Swedish Ethical Review Act, and the Public Access to Information and Secrecy Act, these types of sensitive data can only be made available for specific purposes, including research, that meet the criteria for access to these type of sensitive and confidential data as determined by a legal review. Readers may contact Professor Kristina Alexanderson (kristina.alexanderson@ki.se) regarding the data.

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
