# Peer review of "Occupational Branch and Labor Market Marginalization among Young Employees with Adult Onset of Attention Deficit Hyperactivity Disorder—A Population-Based Matched Cohort Study"

_ijerph, 2022, doi:10.3390/ijerph19127254_

Round 1
Reviewer 1 Report
Interesting study reporting on secondary data analysis of large population-based dataset examining impact of having a diagnosis of ADHD on presence/engagement in the workforce.
p. 1, line 39-40: I believe the correct terminology for the category under which ADHD falls is “Neurodevelopmental Disorder” and should be used instead of “Neuropsychiatric functional disorder”.
Method section => Sample:
=> add the mean and SD for sample chronological age.
This study rests importantly on the diagnostic status of the sample. The authors’ cite the ICD-10 diagnostic category for ADHD - see p. 2, lines 77-79: “We identified individuals with a first diagnosis of ADHD, defined by the code F90, according to the International Classification of Diseases, 10th Revision (ICD-10)(20).” Missing from this section is exactly how the authors ascertained the diagnosis of ADHD in their study sample.
Some statements that were unclear to me included: “In this study, individuals were already employed when diagnosed with ADHD…” (see p. 8; lines 270-271). It was not clear to me as I reviewed the manuscript the data point that denoted the age at which the diagnosis of ADHD was made. How did the authors determine that participants were already employed prior to dx?
Another assertion made on p. 8 that warrants support is “which indicates that they might have been in better health, have less deliberating ADHD symptoms.” Was there a measure of symptom severity? Also, the comorbidity data presented in Table 1 seems to indicate that for the sample with ADHD, ADHD did not often come alone (e.g., 34% comorbid depression/bipolar dx, 38% had comorbid anxiety dx, 20% had comorbid substance use dx….)
Limitation – why not pull out a subsample of ADHD participants who had ADHD but without comorbid mental illness (e.g., depression, anxiety, substance use disorder, other….) and compare them as a 3rd group of ADHD-only?
Author Response
Answer to reviewer 1
We are also grateful for the reviewer’s constructive comments. Please find our detailed responses to the reviewer’s comments below, also indicating the exact text and added references which have been changed in the manuscript.
”p. 1, line 39-40: I believe the correct terminology for the category under which ADHD falls is “Neurodevelopmental Disorder” and should be used instead of “Neuropsychiatric functional disorder”.”
Our answer: we corrected “Neuropsychiatric functional disorder” to “Neurodevelopmental Disorder” in the text
“Method section => Sample:
=>add the mean and SD for sample chronological age.”
Our answer: now we added the age with SD to table 1 for the total sample (24.3, SD:3.0) and for those with and without the ADHD diagnosis (24.2, SD: 3.0 and 24.2, SD: 3.0, respectively).
“This study rests importantly on the diagnostic status of the sample.The authors’ cite the ICD-10 diagnostic category for ADHD - see p. 2, lines 77-79: “We identified individuals with a first diagnosis of ADHD, defined by the code F90, according to the International Classification of Diseases, 10th Revision (ICD-10)(20).” Missing from this section is exactly how the authors ascertained the diagnosis of ADHD in their study sample.”
Our answer: now we added this information as the following sentence to the method section: “and had a health care record with this diagnosis in specialized health care. In Sweden ADHD which has severe enough symptoms to life quality are diagnosed exclusively in specialized healthcare centres using standardized interview (REF: Utredning och diagnostik av ADHD hos vuxna. Brosyr av Socialstyrelsen, Läkemedelsverket, Tandvårds- och läkemedelsförmånsverket, Statens beredning för medicinsk utvärdering, Folkhälsomyndigheten [Internet]. 2014 [cited 2021 Nov 25]. Available from: https://www.socialstyrelsen.se/globalassets/sharepoint-dokument/artikelkatalog/kunskapsstod/2014-10-35.pdf). ”
“Some statements that were unclear to me included: “In this study, individuals were already employed when diagnosed with ADHD…” (see p. 8; lines 270-271). It was not clear to me as I reviewed the manuscript the data point that denoted the age at which the diagnosis of ADHD was made. How did the authors determine that participants were already employed prior to dx?
Our answer: Information on employment was referred to the year prior to the cohort entry. The cohort entry year was equal to the year when ADHD diagnosis was recorded first in the registers. However, we cannot rule out that symptoms of ADHD have already influenced the quality of life before the diagnosis, and there is also some time lag between contacting the health care with symptoms and being referred and get diagnosed in specialized health care centres. As this can considered a limitation to our study we now added the following sentence to the limitation section: “Finally, while information on employment was referred to the year before cohort entry to ensure that all study participant was employed before their ADHD diagnosis, we cannot rule out the symptoms of ADHD has already influenced the participants working life before the diagnosis.”
Another assertion made on p. 8 that warrants support is “which indicates that they might have been in better health, have less deliberating ADHD symptoms.” Was there a measure of symptom severity? Also, the comorbidity data presented in Table 1 seems to indicate that for the sample with ADHD, ADHD did not often come alone (e.g., 34% comorbid depression/bipolar dx, 38% had comorbid anxiety dx, 20% had comorbid substance use dx….)
Our answer: Unfortunately, we had no information on ADHD symptoms severity only accompanied comorbidities. However, in our previous study we showed that both long-term unemployment and mental comorbidities were strong predictors for disability pension among young adults with ADHD. (Chen L, Mittendorfer-Rutz E, Björkenstam E, Rahman S, Gustafsson K, Taipale H, Tanskanen A, Ekselius L, Helgesson M. Risk Factors for Disability Pension among Young Adults Diagnosed with Attention-deficit Hyperactivity Disorder (ADHD) in Adulthood. J Atten Disord. 2022 Mar;26(5):723-734. doi: 10.1177/10870547211025605. Epub 2021 Jun 22. PMID: 34154443; PMCID: PMC8785279.) Now we added this reference to the sentence.
“Limitation – why not pull out a subsample of ADHD participants who had ADHD but without comorbid mental illness (e.g., depression, anxiety, substance use disorder, other….) and compare them as a 3rdgroup of ADHD-only?”
Our answer: We now added a supplement analysis, where we compared the labour market marginalization outcome of those with ADHD with any comorbid mental disorders (n=) of those with ADHD but without any comorbid mental disorders (n=), both stratified by occupational branches and also with adding occupational branch as a covariate to the model. It is now presented in the Supplement material as Supplement table 6. Although we referred to these analyses in the Statistical analysis paragraph and in the Result section as following: “Finally, w stratified the analyses by individuals with ADHD diagnosis with and without any mental comorbidities while adjusting to occupational branches and other covariates.” and “The results comparing individuals with ADHD diagnosis with and without any mental comorbidities (n=4045 and n=1985, respectively) are presented in Supplement Table 5. The probability of LMM was higher among those who had both ADHD and mental comorbidity (OR 4.5, 95% CI: 4.1, 5.0) then those who only were diagnosed by ADHD without other mental comorbidities (OR 2.3, 95% CI: 2.0, 2.5) compared them to the reference group free from ADHD. However, the associations between the different occupational branches were similar in the two subgroups of ADHD (with and without mental comorbidities) (Supplement Table 6)”.
Reviewer 2 Report
It's my pleasure to review your manuscript.
In your manuscript, you play your eyes' on occupational branch and labor-market marginalization among young employees with adult onset of attention deficit hyperactivity disorder in Sweden. There is an important topic in Epidemiology of Occupational Health.
There are some minor comments.
First, in Section Introduction, you should offer more detasils about your contribution.
Second, the theoritical framework and methods should be described with sufficient detail to allow others to replicate and build on published results.
Third, the section number of Conclusion is "5", not "1".
Author Response
Answer to reviewer 2
We are also grateful for the reviewer’s constructive comments. Please find our detailed responses to the reviewer’s comments below, also indicating the exact text and added references which have been changed in the manuscript.
“It's my pleasure to review your manuscript.
In your manuscript, you play your eyes' on occupational branch and labor-market marginalization among young employees with adult onset of attention deficit hyperactivity disorder in Sweden. There is an important topic in Epidemiology of Occupational Health.
There are some minor comments.
First, in Section Introduction, you should offer more detasils about your contribution. Second, the theoritical framework and methods should be described with sufficient detail to allow others to replicate and build on published results.”
Our answer: Thank you very much for the reviewer’s kind words. Now we added some sentences to the Introduction and explain more thoroughly our contribution and the theoretical framework our study was implemented “According to the “core-peripheral” theoretical framework of labour market integration there is a continuum from attachment to marginalization (Atkinson, J. (1984). Manpower strategies for flexible organisations. Personell Management, 16(8), 28-31.; Gustafsson, K., Aronsson, G., Marklund, S., Wikman, A., & Floderus, B. (2014). Peripheral labour market position and risk of disability pension: a prospective population-based study. BMJ open, 4(8), e005230. doi:10.1136/bmjopen-2014-005230). Our previous research show that large proportion of young adults with ADHD diagnosis is already on the periphery of the labour market and has marginalized early (Helgesson M, Björkenstam E, Rahman S, Gustafsson K, Taipale H, Tanskanen A, et al. Labour market marginalisation in young adults diagnosed with attention-deficit hyperactivity disorder (ADHD): a population-based longitudinal cohort study in Sweden. Psychol Med. 2021/07/15 ed. 2021;1–9.). This study aims to understand in what extent labour market related factors, such as occupational branch might play role in LMM among young individuals with ADHD, who has managed to establish labour market attachment”. We also added the STROBE checklist to the supplement material for reporting cohort studies and added some further details to the Methods section on the analytical steps, which we think are sufficient for the replicability of our analysis (page 2-4).
Third, the section number of Conclusion is "5", not "1".
Our answer: we corrected it
Reviewer 3 Report
The paper regards a topic of interest in public health and, to date, under-investigated from an occupational health perspective.
In my opinion, the article needs only a few adjustments. Some observations are reported below.
In the Methods paragraph, please better define the study type. The research would appear to be a retrospective case-control study, but this should be clarified in the text. And, for more precision, the data reporting approach should be explained (which guideline/checklist was followed according to the type of study).
Please move table 1, and its text referrals, to the “Results” paragraph.
There is a lack of reference to “number” and “date” approval by ethics, both in the methods paragraph and at the end of the document in the dedicated section called "Institutional Review Board Statement."
A final remark concerns self-citations. Four papers by the same authors in the references are not a few. Consider whether it is indispensable and proper to cite them all.
I have nothing more to add.
Author Response
Answer to reviewer 3
We are also grateful for the reviewer’s constructive comments. Please find our detailed responses to the reviewer’s comments below, also indicating the exact text and added references which have been changed in the manuscript.
“The paper regards a topic of interest in public health and, to date, under-investigated from an occupational health perspective.
In my opinion, the article needs only a few adjustments. Some observations are reported below.
In the Methods paragraph, please better define the study type. The research would appear to be a retrospective case-control study, but this should be clarified in the text.”
Our answer: now we defined the study type in Methods paragraph: “We conducted a prospective register-based matched cohort study.” The information in the registers was collected prospectively.
And, for more precision, the data reporting approach should be explained (which guideline/checklist was followed according to the type of study).
Our answer: we followed the STROBE guideline/checklist for reporting cohort studies (von Elm E, Altman DG, Egger M, Pocock SJ, Gotzsche PC, Vandenbroucke JP. The Strengthening the Reporting of Observational Studies in Epidemiology (STROBE) Statement: guidelines for reporting observational studies.) Now we added a sentence and the reference to the Method section: “We followed the STROBE guideline/checklist for reporting the study (Supplement table 1). the STROBE checklist to the supplement material.” Although, we added the study design to the title as it is required by the checklist.
“Please move table 1, and its text referrals, to the “Results” paragraph.”
Our answer: now we added a comment for the proof editors to move table1 after the “Results” subheading.
“There is a lack of reference to “number” and “date” approval by ethics, both in the methods paragraph and at the end of the document in the dedicated section called "Institutional Review Board Statement."
Our answer: we added the missing information for the ethics approval under the “Institutional Review Board Statement”
A final remark concerns self-citations. Four papers by the same authors in the references are not a few. Consider whether it is indispensable and proper to cite them all.
Our answer: we understand the reviewer concerns about self-citation. However, this work is part of a comprehensive research project on labour market marginalization among young adults with ADHD, therefore we strongly build on our previous results in this work and refer to some of our key works in this field. Furthermore, research has been very limited or not available in this field yet, therefore studies that we can compare our findings with are very limited. We aim to minimalize self-citation, therefore removed on paper (namely: Helgesson M, Rahman S, Björkenstam E, Gustafsson K, Amin R, Taipale H, et al. Trajectories of labour market marginalisation among young adults with newly diagnosed attention-deficit/hyperactivity disorder (ADHD). Epidemiol Psychiatr Sci. 2021 Oct 22;30:e67. ) from the reference list, which in some sense contains overlapping results with the other papers.
I have nothing more to add.